# A Systematic Review and Meta-Analysis of Enzyme Replacement Therapy in Late-Onset Pompe Disease

**DOI:** 10.3390/jcm10214828

**Published:** 2021-10-21

**Authors:** Alícia Dorneles Dornelles, Ana Paula Pedroso Junges, Tiago Veiga Pereira, Bárbara Corrêa Krug, Candice Beatriz Treter Gonçalves, Juan Clinton Llerena, Priya Sunil Kishnani, Haliton Alves de Oliveira, Ida Vanessa Doederlein Schwartz

**Affiliations:** 1Postgraduate Program in Medical Sciences, Faculty of Medicine, Universidade Federal do Rio Grande do Sul, Porto Alegre CEP 90035003, Brazil; alidorneles@gmail.com; 2Medical Genetics Service, Hospital de Clínicas de Porto Alegre, Porto Alegre CEP 90035903, Brazil; apjunges@hcpa.edu.br; 3Faculty of Medicine, Universidade Federal do Rio Grande do Sul, Porto Alegre CEP 90035003, Brazil; 4Applied Health Research Centre, Li Ka Shing Knowledge Institute, St Michael’s Hospital, Toronto, ON M5B 1T8, Canada; tiago.pereira@metadatum.com.br; 5Department of Health Sciences, College of Medicine, University of Leicester, Leicester LE1 7RH, UK; 6Nuclimed, Clinical Research Center, Hospital de Clinicas de Porto Alegre, Porto Alegre CEP 90035903, Brazil; krugbarbara@gmail.com (B.C.K.); candicebtg@gmail.com (C.B.T.G.); 7Instituto Fernandes Figueira, Fiocruz, Rio de Janeiro CEP 22250020, Brazil; juan.llerena@iff.fiocruz.br; 8Department of Pediatrics, Duke University Medical Center, Durham, NC 27710, USA; priya.kishnani@duke.edu; 9Health Technology Assessment Unit, Hospital Alemão Oswaldo Cruz, São Paulo CEP 01323903, Brazil; haoliveira@haoc.com.br; 10Department of Genetics, Universidade Federal do Rio Grande do Sul, Porto Alegre CEP 91501970, Brazil

**Keywords:** glycogen storage disease type II, alpha-glucosidase, Pompe disease, enzyme replacement therapy

## Abstract

Pompe disease (PD) is a glycogen storage disorder caused by deficient activity of acid alpha-glucosidase (GAA). We sought to review the latest available evidence on the safety and efficacy of recombinant human GAA enzyme replacement therapy (ERT) for late-onset PD (LOPD). Methods: We systematically searched the MEDLINE (via PubMed), Embase, and Cochrane databases for prospective clinical studies evaluating ERT for LOPD on pre-specified outcomes. A meta-analysis was also performed. Results: Of 1601 articles identified, 22 were included. Studies were heterogeneous and with very low certainty of evidence for most outcomes. The following outcomes showed improvements associated with GAA ERT, over a mean follow-up of 32.5 months: distance walked in the 6-min walking test (6MWT) (mean change 35.7 m (95% confidence interval [CI] 7.78, 63.75)), physical domain of the SF-36 quality of life (QOL) questionnaire (mean change 1.96 (95% CI 0.33, 3.59)), and time on ventilation (TOV) (mean change −2.64 h (95% CI −5.28, 0.00)). There were no differences between the pre- and post-ERT period for functional vital capacity (FVC), Walton and Gardner-Medwin Scale score, upper-limb strength, or total SF-36 QOL score. Adverse events (AEs) after ERT were mild in most cases. Conclusion: Considering the limitations imposed by the rarity of PD, our data suggest that GAA ERT improves 6MWT, physical QOL, and TOV in LOPD patients. ERT was safe in the studied population. PROSPERO register: 135102.

## 1. Introduction

Pompe disease (PD), or type II glycogenosis, is a rare genetic disease characterized by progressive neuromuscular involvement, often fatal in severe forms [1]. It is caused by deficient activity of acid alpha-glucosidase (also known as acid maltase), a lysosomal enzyme encoded by the *GAA* gene that breaks down glycogen into glucose [1]. This deficient activity, caused by biallelic pathogenic variants in *GAA*, leads to lysosomal glycogen accumulation in the skeletal and cardiac muscles, hindering cell function and ultimately destroying cells by hypertrophy and lysosome rupture [1,2,3,4,5,6].

Residual enzyme activity correlates positively with age at disease onset and inversely with the rate of disease progression, which allows PD to be classified according to the age of onset, cardiac involvement, and speed of progression. When clinical onset occurs before the age of 12 months and cardiomyopathy is present, it is known as infantile-onset PD (IOPD); all other forms are referred to as late-onset PD (LOPD) [1,7]. LOPD occurs on a spectrum; patients may present through the second year of life without cardiomyopathy, in childhood, adolescence, or at any point in adult life [1,7]. Although the overall severity of involvement is variable, life expectancy is generally shorter than in healthy individuals [1,2,8].

The overall incidence of PD worldwide is around 1 in 40,000 newborns (NBs). The incidence is higher in African Americans (1/12,000 NBs) and lower in Chinese (1/40,000 to 1/50,000 NBs) individuals [1,2,3]. After the inclusion of PD in the newborn screening programs of some countries, more reliable estimates of its incidence have emerged: 1/26,319 NBs in Illinois (USA) [9], 1/17,134 NBs in Pennsylvania (USA) [10], 1/10,152 NBs in Missouri (USA) [11], and 1/34,402 NBs in the Asian population of Japan [12].

There is no curative treatment for PD. Currently available treatment options are designed to address the mutant protein and consist of enzyme replacement therapy (ERT) with alglucosidase alfa (Myozyme™), a form of human acid alpha-glucosidase (GAA) produced by recombinant DNA technology in Chinese hamster ovary cells [2]. The recommended dosage regimen of alglucosidase alfa is 20 mg/kg body weight, administered every 2 weeks by intravenous (IV) infusion [8].

Two previous systematic reviews aimed to assess the effectiveness, safety, and appropriate dose regimen of enzyme replacement therapy (ERT) for treating LOPD; however, both have limitations in important domains. One, which included a meta-analysis, evaluated survival, vital capacity, and performance in the 6-min walking test (6MWT), reporting improvements in all outcomes [13]. The second, which did not perform a meta-analysis, showed improvement in the 6MWT but failed to include relevant prospective cohort studies [14]. Therefore, the impact of alglucosidase alfa treatment on key outcomes, such as quality of life (QOL) and time on ventilatory support (TOV), is still unclear. A further, significant knowledge gap that still remains is the ideal timing of ERT initiation. Within this context, the present systematic review with meta-analysis was designed to evaluate the effects of alglucosidase alfa ERT in LOPD.

## 2. Methodology

This study aimed to review the latest available evidence on the effects of alglucosidase alfa ERT in LOPD and its safety. To guide the literature search, a structured PICO question was formulated as follows: “Is the use of alglucosidase alfa as effective and safe as ERT in patients with PD?”. The systematic review is reported as proposed by the PRISMA Guidelines [15] and has been registered in the PROSPERO database (123700).

### 2.1. Information Sources and Search Strategy

The MEDLINE (via PubMed), Embase, and Cochrane Central Register of Controlled Trials databases were searched for studies published before 30 May 2021. The search strategies are shown in Table 1; for the Cochrane Library, we used both strategies combined.

### 2.2. Eligibility Criteria and Study Selection

We planned to include only randomized clinical trials (RCT) and observational comparative studies in which ERT with alglucosidase alfa was used for the treatment of patients with LOPD. Other prospective study designs were included (open-label and non-randomized trials, controlled or otherwise, including quasi-experimental designs) if the sample size was ≥5. In vitro studies or animal models, reviews, expert opinions, and retrospective studies were excluded. Unpublished work was covered by the identification of conference abstracts containing data deemed to be of interest. The final published articles were then included when available.

Studies that did not evaluate at least one of the eight outcomes of interest, defined a priori by a team of experts, were excluded. These outcomes were QOL, functional capacity (6MWT, forced vital capacity (FVC), and Walton and Gardner-Medwin Scale (WGMS) score), survival, TOV in hours/day, muscle strength, sleep quality, swallowing, and safety. For FVC, an increase of at least 10% after the intervention was considered a clinically relevant improvement [16,17]. For 6MWT, an increase of at least 26 m was considered a clinically significant change, as recommended by Schrover et al. for muscular diseases [18]. The WGMS is a scale that evaluates functional activity on a point system ranging from 0 = normal to 10 = bedridden.

The selection stage was performed independently by two investigators (APPJ, CG), who assessed the abstracts retrieved during the search for eligibility. Decisions were compared, and articles deemed relevant were forwarded to two other investigators (ADD, BK) who, independently, using standardized data collection forms, extracted information on the characteristics of these studies (design, randomization methods, population of participants, interventions, and outcomes). The two investigators then took part in a consensus meeting. Any disagreement that remained was addressed by the intervention of a third investigator (IVDS). Finally, the references of the selected articles were hand-searched for potentially relevant studies not identified by the previous search strategies. When such information could not be retrieved, an email was sent to authors requesting non-reported data.

### 2.3. Data Collection

Studies with overlapping data were excluded from meta-analyses. In these cases, the study with the largest sample (or, if both studies had the same sample size, that with the longest follow-up) was retained for analysis.

### 2.4. Statistical Analysis

We summarized results using mean changes from baseline with 95% confidence intervals (CIs) for continuous outcomes. To incorporate follow-up time, we used incidence rates (IRs) with 95% CIs to summarize events. To facilitate interpretation, we standardized all IR estimates in events per 100 person-years. Study-specific mean changes were combined through an inverse-variance random-effects model with the restricted maximum-likelihood estimator (REML) of between-study variance (τ 2) for continuous variables. Events were combined with a generalized linear mixed model (GLMM) [19], in which a random intercept logistic regression model was fitted (log transformation) with a maximum-likelihood (ML) estimator. Sparse data were naturally taken into account in the GLMM, and no continuity corrections were used. The random-effects model was used for the primary analysis, but summary estimates obtained with a fixed-effects model (inverse-variance) were presented as a sensitivity analysis.

When not directly reported, mean change from baseline and standard error estimates were approximated based on reported statistics (95% CI, *p*-values, median, and interquartile range). We imputed standard deviations for baseline changes, assuming a correlation of 0.7 between baseline and follow-up scores. When only the median and interquartile range were informed, we used an approximate Bayesian computation (ABC) model to estimate means and standard deviations [20]. We employed clinically plausible ranges for the prior [~uniform (0,100)] distributions derived from studies reporting complete information and the opinion of specialists. Statistical heterogeneity was tested with Cochran’s Q test and quantified with the I^2^ metric. Cochran’s Q was considered statistically significant for heterogeneity if *p* < 0.10 [21]. No threshold for statistical significance was used for the evaluation of clinical variables. Analyses were performed with Stata (version 16, StataCorp, College Station, TX, USA) and R (version 3.2.3, R Core Team, The R Foundation, Vienna, Austria).

### 2.5. Evaluation of the Quality of Included Studies

The quality of included studies was evaluated with tools appropriate for the study designs: Risk of Bias tool (RoB) 2.0 for RCTs and Risk Of Bias In Non-randomized Studies of Interventions tool (ROBINS-I) for non-randomized studies of interventions (NRSI) [22,23]. Certainty of evidence of outcomes defined a priori was evaluated according to GRADE criteria [24,25,26]. Assessment of certainty of evidence for outcomes was performed independently by two investigators (ADD, HAOJ).

## 3. Results

The broad search strategy retrieved 1601 references (768 from MEDLINE, 833 from EMBASE, none from the Cochrane Library), of which 242 were duplicated. The titles and abstracts of 1359 references were read, and 33 publications were selected for full-text evaluation. Of these, 22 were selected for eligibility, and 11 were excluded. A flow diagram of evidence selection is shown in Figure 1. Ultimately, 22 studies were identified for LOPD, including one RCT.

Studies that evaluated the outcomes defined a priori are described in Table 2. Our search did not retrieve any articles evaluating sleep quality, survival, or swallowing disorder that matched the inclusion criteria; therefore, these outcomes could not be evaluated.

### 3.1. Characteristics of Included Studies

All included studies and their characteristics are described in Table 3. Only one double-blind RCT was identified and included [36].

### 3.2. Assessment of Functional Capacity

#### 3.2.1. Forced Vital Capacity

Based on data from 15 studies (participants = 348; mean follow-up = 36.8 mo; Table 2 [13,27,28,29,30,31,32,33,34,35,36,37,38,39,40]), there was no evidence of improvement in FVC during the performance of spirometry in the sitting, supine, or orthostatic positions (within-group mean change: 0.41% (95% CI: −0.3 to 1.12%)), as shown in Figure 2 and Appendix A. There was low heterogeneity between studies and very low certainty of evidence (Appendix A).

#### 3.2.2. Six-Minute Walking Test

Performance on the 6MWT was evaluated in 14 studies before and after ERT (participants = 348; mean follow-up = 36.8 mo), as shown in Table 2 [13,27,28,29,30,31,34,35,36,37,38,39,41,42] and Appendix A. Van der Ploeg et al. (2010) [36] included the same population as van der Ploeg et al. (2012) [29] and was thus excluded; Kuperus et al. [39] measured the outcome as median and, therefore, was excluded from meta-analysis as well. Despite the considerable heterogeneity between studies and very low certainty of evidence (Appendix A), there was evidence of clinically significant improvement after treatment (within-group mean change: 35.7 m (95% CI: 7.78 to 63.75); Figure 3).

#### 3.2.3. Walton and Gardner-Medwin Scale (WGMS)

Six studies evaluated WGMS scores, as shown in Table 2 [13,27,30,31,40,43] (results shown in Appendix A). The data suggest that ERT had no effect on the WGMS score in any of the included studies. Certainty of evidence was very low (Appendix A), mainly due to the imprecision of the results of the included studies.

### 3.3. Upper-Limb Strength

Nine studies evaluated strength in the upper limbs, but did so very heterogeneously (Appendix A). Although several studies carried out strength assessment according to the Medical Research Council (MRC) scale [13,30,33,46], they reported different methods of calculating it, evaluated different muscle groups, and some did not evaluate upper and lower limbs separately, limiting the comparability of this outcome. Thereby, muscle strength was evaluated through a meta-analysis in relation to two variables: handheld dynamometry and the Quick Motor Function Test. Handheld dynamometry was evaluated in 2 of 9 studies, without significant differences (mean change 244.05 (95% CI −151.18, 639.27)) (Figure 4), with considerable heterogeneity between studies; the Quick Motor Function Test was evaluated in 3 of 9 studies, without a significant difference (mean change 7.85 (95% CI −2.48, 18.18)) (Figure 5), with substantial heterogeneity between studies and very low certainty of evidence (Appendix A).

### 3.4. Quality of Life

Six studies evaluated this outcome; the meta-analysis of their results is shown in Figure 6 and detailed in Appendix A. The instrument most commonly used to assess QOL was the Medical Outcome Study 36-item Short Form Health Survey, or SF-36 questionnaire [13,30,36,43,47]. The SF-36 is a generic instrument that has been widely used, has been translated into many languages, and has been shown to have good reliability and validity. Strothotte et al. [13] was excluded from the meta-analysis due to incomplete data available, and van der Ploeg et al. (2016) was excluded due to the use of the Pediatric Quality of Life Inventory™ (PedsQL) [28].

There were no differences in overall QOL (mean change 7.05 (95% CI −7.30, 21.41)) or in the mental component of the SF-36 (mean change 5.37 (95% CI −4.04, 14.78)), with considerable heterogeneity. However, there was a difference in the physical component (mean change 1.96 (95% CI 0.33, 3.59)) (Figure 6), with substantial heterogeneity and very low certainty of evidence (Appendix A).

### 3.5. Time on Ventilation

Six studies evaluated this outcome in LOPD; a synthesis of their results is shown in Figure 7 and Appendix A. Two studies [30,33] were not included in the meta-analysis due to a lack of data. There was weak evidence indicating that ERT, on average, is associated with a positive effect on TOV, despite substantial heterogeneity between studies and low certainty of evidence (Appendix A) (mean change −2.64 h (95% CI −5.28, 0.00)).

### 3.6. Safety

#### 3.6.1. Adverse Events

Several studies have assessed the safety profile of ERT concerning the presence of adverse events (AEs) and infusion-associated reactions (IARs), as shown in Table 4 and Appendix A [13,28,29,30,31,33,34,36,39,40,42,43,44,45]. The reported AEs are tachycardia, desaturation, malaise, chills, facial erythema, erythema at the enzyme infusion site, urticarial reactions, hyperhidrosis, chest discomfort, vomiting, systemic arterial hypertension, flu-like symptoms, pruritus, bronchospasm, and hyperthermia. Certainty of evidence is presented in Appendix A.

In the study by van der Ploeg et al. (2016) [28], there was only one severe AE (not specified), unrelated to treatment. Mild and moderate AEs occurred in 35.5% of patients, with 25% experiencing IARs. The incidence of IARs reported by de Vries et al. (2017) [45] was 18% (13/73 patients), the most common being malaise, chills, and hyperthermia. In another study by the same author [33], 12/69 patients (17%) developed an infusion reaction; however, only three patients remained symptomatic after administration of antihistamines and corticosteroids.

Angelini et al. (2012) [31] reported the following adverse reactions to ERT, occurring in 4/74 patients (6%) and considered of moderate intensity: facial erythema, erythema at the infusion site, flu-like symptoms, generalized pruritus, and bronchospasm (also described by Bembi et al. [42]). The symptoms were controlled with antihistamines. In another study by Orlikowski et al. with five patients, 58 mild to moderate AEs were described after starting treatment, including erythema and hyperthermia.

In the LOTS RCT, the ERT group and the placebo group had similar frequencies of severe AEs, treatment-related events, and infusion reactions (Appendix A). The treatment group had a higher frequency of mild to moderate AEs, which did not prevent the continuation of treatment. Urticariform reactions (also described by Bembi et al. [42]), hyperhidrosis, chest discomfort, flushing, vomiting, and increased blood pressure occurred in up to 8% of patients treated with alglucosidase alfa, and were not reported in the placebo group. These findings were corroborated by an extension study with the same population (LOTS Extension) [29]. In addition, Forsha et al. [44], using data obtained in the LOTS study, evaluated only the safety of ERT regarding cardiovascular events that occurred after the initiation of alglucosidase alfa, and found no significant difference between the treatment and placebo groups in change in ejection fraction (*p* = 0.8), PR interval (*p* = 0.71), ventricular mass (*p* = 0.71), or QRS duration (*p* = 0.67).

In the study by Kuperus et al. [39], 19 patients (22%) had at least one AE, all of which were controlled by a reduced infusion rate or premedication (antihistamines or corticosteroids). ERT was discontinued in four patients, but in only one for safety reasons (a patient with a history of autoimmune diseases and drug allergies before treatment developed multiple IARs). AEs have also been described in the study by Strothotte et al. [13], not included in meta-analysis due to incomplete data, and by Regnery et al. [30]: erythema, tachycardia, desaturation, rash, and pruritus, with no deaths occurring during the 12 and 36 months, respectively, of these studies. In the study by Vianello et al. [40], no AE has been described. In the study by van Capelle et al. [34], five patients were treated with ERT for 3 years and no AEs were observed in any of the 390 total intravenous infusions.

#### 3.6.2. Mortality

Data on mortality were described in nine studies (Table 4 and Appendix A); however, only five described the occurrence of the event [31,33,36,39,43]. In the study conducted by Orlikowski et al. [43] with five patients, there was one death due to tracheal hemorrhage not related to treatment. In the study with the highest overall mortality rate, Kuperus et al. [39], 1/19 patients died of respiratory failure at 56 years of age, 1.1 years after discontinuing ERT for personal reasons. Another 6/19 patients on ERT died, though no deaths were considered treatment-related. The incidence rate of death (events per person-year) across all included studies was 4.4 events per 1000 person-years (95% CI 1.5 to 12.8).

#### 3.6.3. Anti-Alglucosidase Alfa Antibodies

Data on anti-alglucosidase alfa antibody (Ab) titers for patients receiving ERT are shown in Table 5. Although most patients presented with elevated Ab titers, few showed a reduction in response to treatment or higher incidence of AEs. The most in-depth evaluation of Ab titers for patients receiving ERT was by de Vries et al. (2017) [45], and this study will therefore be described in greater detail. The patients were divided into three groups, according to their respective Ab titers: the first, of 16 patients, corresponded to high titers (>1: 31,250); the second, of 29 patients, corresponded to moderate titers (1: 1250 to <1: 31,250); the third, of 28 patients, to low titers (0 to <1: 1250). Three patterns of progression were observed concerning anti-ERT Ab titers; in the vast majority of patients (97%), titers either decreased or remained stable after 12 months of treatment, except in two patients, one of whom belonged to the group with the highest titers and the other to intermediate titers. Using the combined score of the MRC scale and the standing FVC, the authors compared treatment responses between the three Ab titer groups and found no significant differences between them, whether at baseline or after 3 years of treatment (*p* = 0.35 and *p* = 0.38, respectively). In one patient with high titers, the Ab had neutralizing effects on the enzyme, with a decline in FVC and strength. De Vries et al. (2017) [45] concluded, therefore, that there was a relationship between anti-alglucosidase alfa Ab titers and the development of AEs (statistical analysis not shown). Only 1/28 (4%) patients in the low-titer group had AEs, while 5/29 (17%) in the intermediate-titer group and 7/16 (44%) in the high-titer group experienced them.

### 3.7. Risk of Bias and Quality of Included Studies

The risk of bias of the included NRSI is shown in Figure 8. The RCT included had some concerning issues, such as an imbalance in age at baseline and possible conflict of interest of the investigators. Most included articles showed a moderate to severe risk of bias, independently of the study design; only one showed a critical risk of bias [46].

### 3.8. Certainty of Evidence by Outcomes

All outcomes were assessed for certainty of evidence, with low certainty only for TOV. All other outcomes had very low certainty of evidence, mainly due to the uncontrolled observational design of the included studies, with data from secondary outcomes. A full analysis is available in the Appendix A.

## 4. Discussion

LOPD is a rare, serious disease with no specific treatment available other than ERT. Interpretation of the available evidence must always consider these facts. Given that only one double-blind RCT of GAA ERT for LOPD has been conducted [36], prospective observational trials were also evaluated in this review. The included studies all had small sample sizes (which is to be expected given the rarity of LOPD), as well as different ages (children, adolescents, and adults), stages, durations of disease burden prior to start of ERT, and phenotypic manifestations. Among the outcomes evaluated for LOPD, we showed a benefit for 6MWT, as also demonstrated in a previous meta-analysis [50]. Otherwise, this was the first meta-analysis to evaluate the effect of ERT on TOV and QOL. An improvement in functional capacity, measured through FVC, was not confirmed.

Two systematic reviews have already been published on the efficacy of alglucosidase alfa, neither of which assessed safety directly [14,50]: the first one in 2013, by Toscano et al. [14], and the second one in 2017, by Schoser et al. [50]. The latter included 22 papers (of which 10 were included in the present review), comprising case series and retrospective studies, with or without a control group, and evaluated 6MWT, FVC, and mortality by conducting a meta-analysis. The average age of patients included in the review was 46 years. [50] Toscano et al. included 21 papers (of which 14 were included in the present review), also comprising case series and retrospective studies, with or without a control group. They evaluated, among other outcomes, 6MWT, FVC, the need for ventilatory support, and QOL, with most patients aged between 40 and 59 years (44%); any improvement was considered meaningful, without defined criteria for clinical relevance [14]. Our review included 22 papers only with prospective data, excluding case series, and the mean age of the included participants was 42.8 years (range, 7 to 76.3 years), with 32.5 months of follow-up.

The systematic review carried out by Toscano et al. [14] evaluated FVC for 124 treated patients and identified an improvement in 51.6%, stable disease in 13.7%, and decline in 34.7%. There was no correlation between the duration of treatment and improvement in lung capacity, with no description of comparison with a control group. Likewise, the meta-analysis carried out by Schoser et al. [50] demonstrated a beneficial effect of ERT on FVC in groups receiving alglucosidase alfa; their conclusion was based on an analysis using a fixed-effect model, which is not considered the best method of evaluating studies with heterogeneous populations. The results showed that untreated patients had a 2.3% decline in FVC% after 12 months and 6.2% after 4 years; treated patients had an initial increase in FVC% of 1.4% after 2 months, with a return to baseline FVC% and a slight decline in follow-up, with data not detailed. The difference in efficacy between control (a historical cohort without treatment) and treated patients varied from 4.5% after 1 year to 6% at 4 years, and therefore cannot be considered clinically significant. Our data, retrieved from 348 patients, do not confirm the findings of Schoser et al., suggesting that there is little effect of ERT on FVC.

Our results confirm a previously published meta-analysis concerning the effect of ERT on 6MWT (mean change 36.6 m (95% CI 10.72, 62.48)). Schoser et al. [50] demonstrated a beneficial effect of ERT on 6MWT in the groups receiving alglucosidase alfa at 12 months, with an average improvement 43 m greater than in controls, based on data from 171 patients undergoing treatment in comparison to a historical cohort without treatment. Toscano et al. [14] also included the 6MWT among their outcomes of interest and reported data from 122 patients, of which 77.9% improved, 8.2% stabilized, and 13.9% worsened. We found an increase of 36 m in this outcome, above the 26 m cutoff considered clinically relevant, and the improvement in walking distance described in the included articles ranged from 33 to 1000 m. The inclusion of the results from Bembi et al. (2010) [42] for this outcome highlights the issue of early intervention, as the greatest benefit of intervention was seen in the young group, which is of fundamental importance for the analysis of the available evidence, since additional benefits or a greater effect size may not have been found in other studies due to the inclusion of patients with established disease and very heterogeneous age at initiation of treatment.

Our study also indicates a beneficial effect of ERT on the physical component of QOL (mean change 1.96 (95% CI 0.33, 3.59)), and this improvement in endurance has positive aspects in making patients less dependent on caregivers. However, it bears stressing that generic QOL instruments were used, as there is no specific validated instrument for assessing the QOL of patients with PD. One systematic review without meta-analysis also evaluated QOL [14], and all included studies in this systematic review with *n* ≥ 5 were included in our search. In addition, 9 of the 21 included articles evaluated QOL, with 156 patients evaluated for this outcome using the SF-36 questionnaire. Qualitative synthesis of these studies showed that only 13/156 patients (8.3%) improved their QOL scores after treatment.

This is the first meta-analysis to show a trend toward a beneficial effect of ERT on TOV (mean change −2.64 h (95% CI −5.28, 0.00)), although previous systematic reviews have evaluated this outcome in different ways. In Schoser et al. [50], the percentage of patients dependent on ventilation varied between 14 and 100%, which is considered very high, due to the early age of the sample. They also reported that the number of patients requiring ventilation was maintained in patients treated with ERT, compared to a historical cohort that presented an increased proportion of patients using ventilation (data not provided by Schoser et al.)—with great heterogeneity between studies, however. Toscano et al. [14] evaluated data from 66 patients and demonstrated that ERT resulted in an improvement in the need for MV in 59.1% of patients, stabilization in 36.4% of patients, and worsening in only 4.5% of patients. Regarding the need for non-invasive ventilation, 64.1% improved, 32.1% stabilized, and 3.8% worsened (data not detailed). In our study, 265 patients were on any type of ventilatory support.

Concerning mortality, our study showed an incidence rate of 4.4/1000 events per person-year of patients with LOPD treated with ERT, which is considered very low. A previous systematic review with meta-analysis [50] was able to estimate that patients on ERT had a lower mortality rate compared to those not treated, using mortality data from included studies. It is important to note that the results of this study should be interpreted with caution, as the methodological steps followed by authors were not clearly mentioned.

Schoser et al. included six observational studies [31,32,43,46,51,52], of which only one [51] presented comparative data between a group on alglucosidase alfa ERT and untreated controls. According to this meta-analysis, the summary measure showed a 79% reduction in risk of death for those patients on alglucosidase alfa (HR = 0.21 (95% CI 0.11; 0.41)). However, it should be noted that the other studies, which were non-comparative case series, might have led to an overestimation of the effect. Although this study corrected the meta-analysis by covariables (meta-regression adjusted for age, sex, and severity of PD) and interpreted the sources of heterogeneity, the review had a serious risk of bias, according to the AMSTAR-2 tool. Gungor et al. (2013) [51] showed a 59% reduced risk of death for those patients on alglucosidase alfa compared to those who did not receive the therapy (HR = 0.41 (95% CI 0.19; 0.87)). This analysis was adjusted for age, sex, country of residence, and severity of PD. The other studies were case series, with no comparator group, reported a small number of deaths, and did not discuss whether these were related to treatment (absolute frequency in the five studies: 3 deaths among 151 patients). Gungor et al. (2013) [51] was the only study with a comparator group and cohort adjusted for potential confounders.

Although treatment-emergent or infusion-related AEs are common in ERT recipients, in most cases, these are mild and easily treatable. The development of IgG antibodies to alglucosidase alfa is also frequent; however, it does not seem to be related to the presence or absence of AEs or to the effect of treatment, although the measures used may not be sensitive enough to identify these. In IOPD, where the outcome is clearer, there is a direct correlation between patients with highly sustained and sustained intermediate Ab titers and clinical outcome. Therefore, more studies in LOPD are needed evaluating outcome measures that have the ability to capture small changes caused by ERT on them [53]. A systematic review conducted by Toscano et al. [14] described AEs as mostly mild to moderate in 303 patients undergoing treatment, with severe AEs reported in only four patients, all in studies included in the present article. They also reported the development of antibodies in 128 patients, three of whom had an anaphylactic reaction. It bears stressing that the evidence for safety is low, but although these studies have already reached a substantial length of follow-up, ERT is generally understood to be safe. No studies have demonstrated, for example, the effectiveness of physical therapy as the sole treatment strategy for these patients.

The included studies were highly heterogeneous, enrolling patients from different age groups and with different disease severities. Therefore, the findings of this review and meta-analysis should be interpreted with caution. In addition, most included studies were non-randomized studies of interventions, with low methodological quality, no comparison group, or comparison with a historical cohort. Another challenge is incompletely reported data, as publication bias was present. Usually, data from secondary outcomes were poorly reported, as studies were not designed to measure them. In most cases, however, conclusions were made without showing full data and results. Proper reporting of data is essential. One strength of the present study was that we included only prospective trials, in an attempt to avoid memory and selection bias. Prospective trials also have the advantage of collecting data accordingly with the predefined outcomes of interest, which does not happen in retrospective trials.

In conclusion, alglucosidase alfa effectively increases the distance achieved in 6MWT, improves the physical component of QOL, and may decrease TOV in treated patients. Stabilization of functional capacity, measured through FVC, was not confirmed. The treatment is safe in the studied population, with generally mild adverse events. Further studies could evaluate the impact of the duration of follow-up in the included studies, taking into consideration that the efficacy of ERT may present some secondary decline after 3 to 5 years of treatment [54]. Moreover, the age at ERT initiation is an important aspect that should be addressed in future reviews.

## Figures and Tables

**Figure 1 jcm-10-04828-f001:**
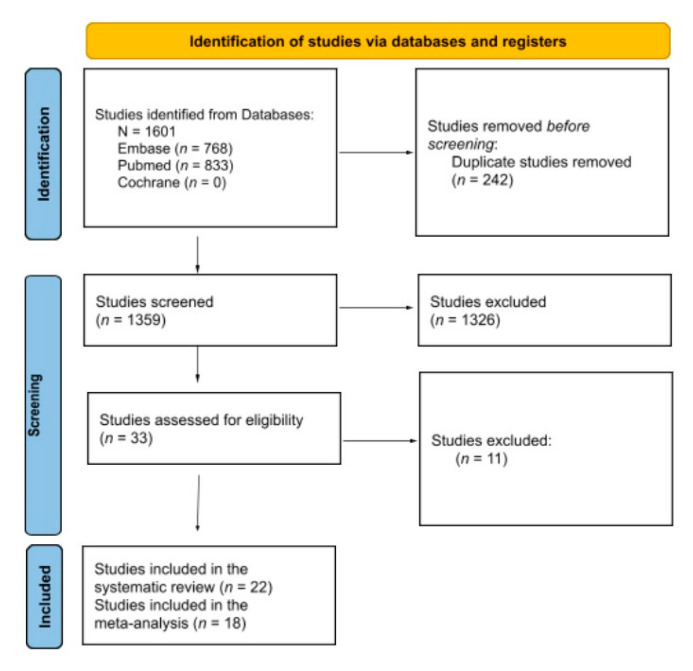
PRISMA flow diagram of search results.

**Figure 2 jcm-10-04828-f002:**
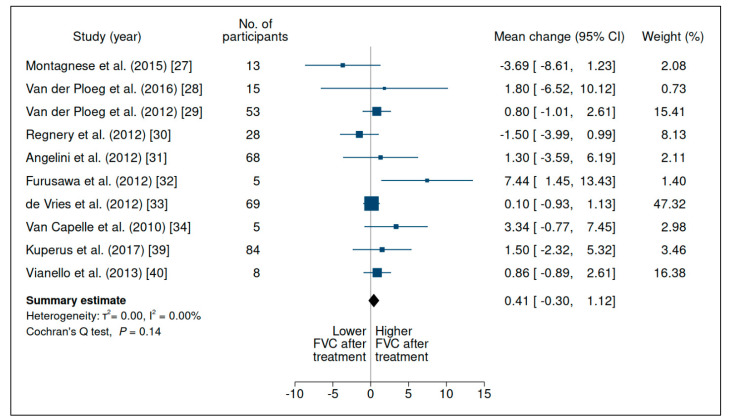
Evaluation of forced vital capacity (% of predicted) in upright position in patients with late-onset Pompe disease on enzyme replacement therapy with alglucosidase alfa. Weights are inverse-variance weights and are proportional to the contribution of each study to the summary estimate. I^2^ is the fraction of variance that is due to statistical heterogeneity and not chance. Tau^2^ denotes the between-study variance.

**Figure 3 jcm-10-04828-f003:**
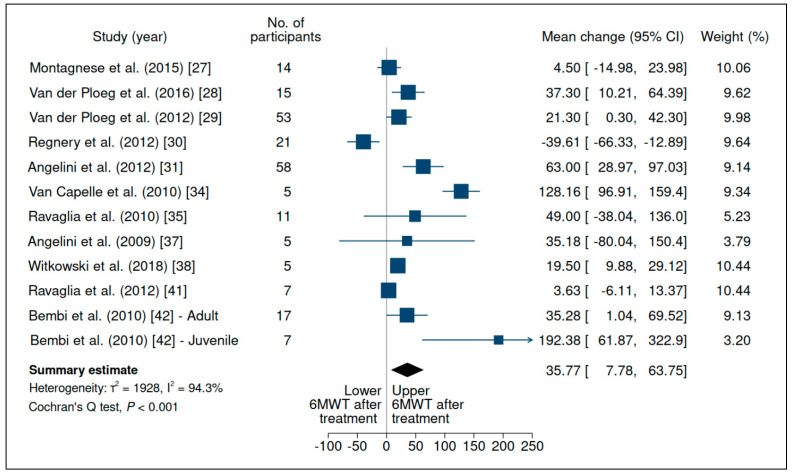
Evaluation of 6-min walking test performance (distance walked in meters) in patients with late-onset Pompe disease on enzyme replacement therapy with alglucosidase alfa. Weights are inverse-variance weights and are proportional to the contribution of each study to the summary estimate. I^2^ is the fraction of variance that is due to statistical heterogeneity and not chance. Tau^2^ denotes the between-study variance.

**Figure 4 jcm-10-04828-f004:**
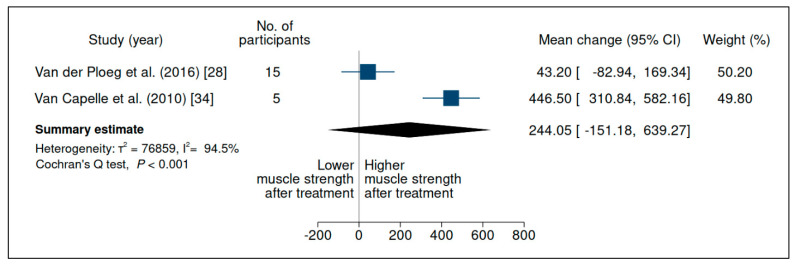
Evaluation of handheld dynamometry in patients with late-onset Pompe disease on enzyme replacement therapy with alglucosidase alfa. Weights are inverse-variance weights and are proportional to the contribution of each study to the summary estimate. I^2^ is the fraction of variance that is due to statistical heterogeneity and not chance. Tau^2^ denotes the between-study variance.

**Figure 5 jcm-10-04828-f005:**
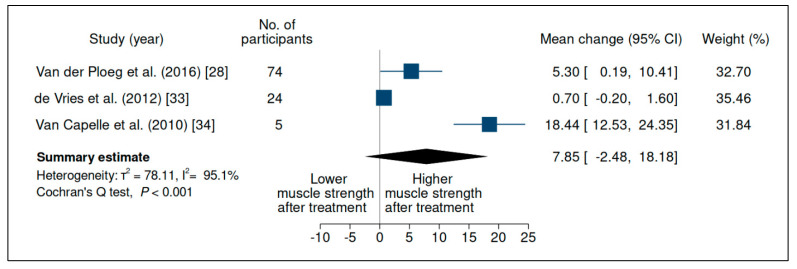
Evaluation of the Quick Motor Function Test in patients with late-onset Pompe disease on enzyme replacement therapy with alglucosidase alfa. Weights are inverse-variance weights and are proportional to the contribution of each study to the summary estimate. I^2^ is the fraction of variance that is due to statistical heterogeneity and not chance. Tau^2^ denotes the between-study variance.

**Figure 6 jcm-10-04828-f006:**
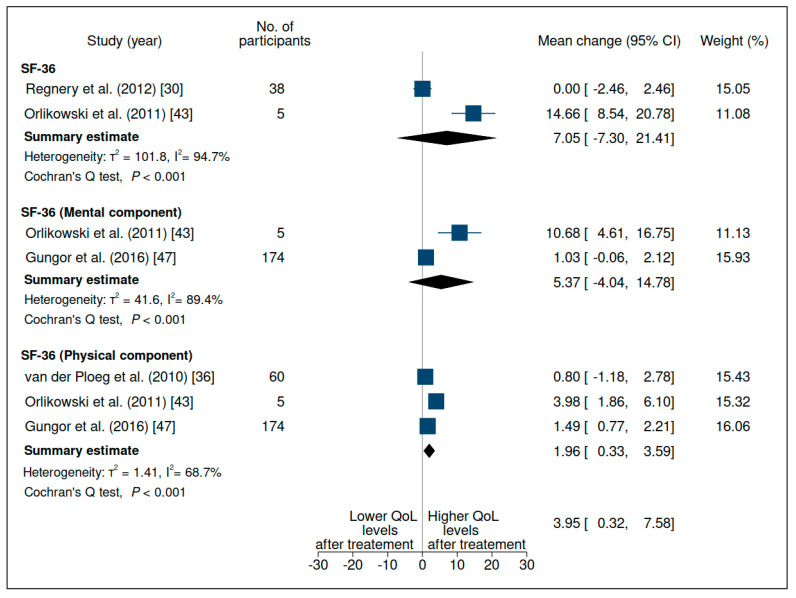
Evaluation of the quality of life of patients with late-onset Pompe disease on enzyme replacement therapy with alglucosidase alfa. Weights are inverse-variance weights and are proportional to the contribution of each study to the summary estimate. I^2^ is the fraction of variance that is due to statistical heterogeneity and not chance. Tau^2^ denotes the between-study variance.

**Figure 7 jcm-10-04828-f007:**
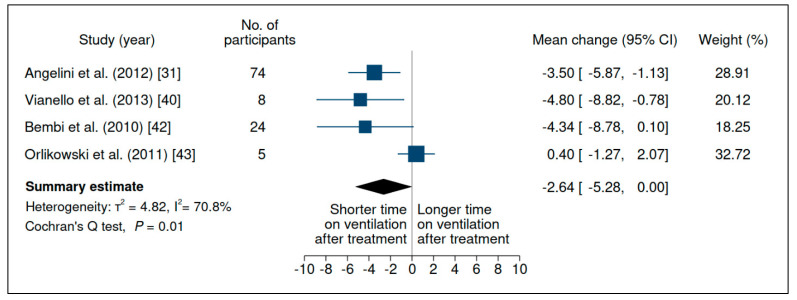
Evaluation of the time on ventilation (h) of patients with late-onset Pompe disease on enzyme replacement therapy with alglucosidase alfa. Weights are inverse-variance weights and are proportional to the contribution of each study to the summary estimate. I^2^ is the fraction of variance that is due to statistical heterogeneity and not chance. Tau^2^ denotes the between-study variance.

**Figure 8 jcm-10-04828-f008:**
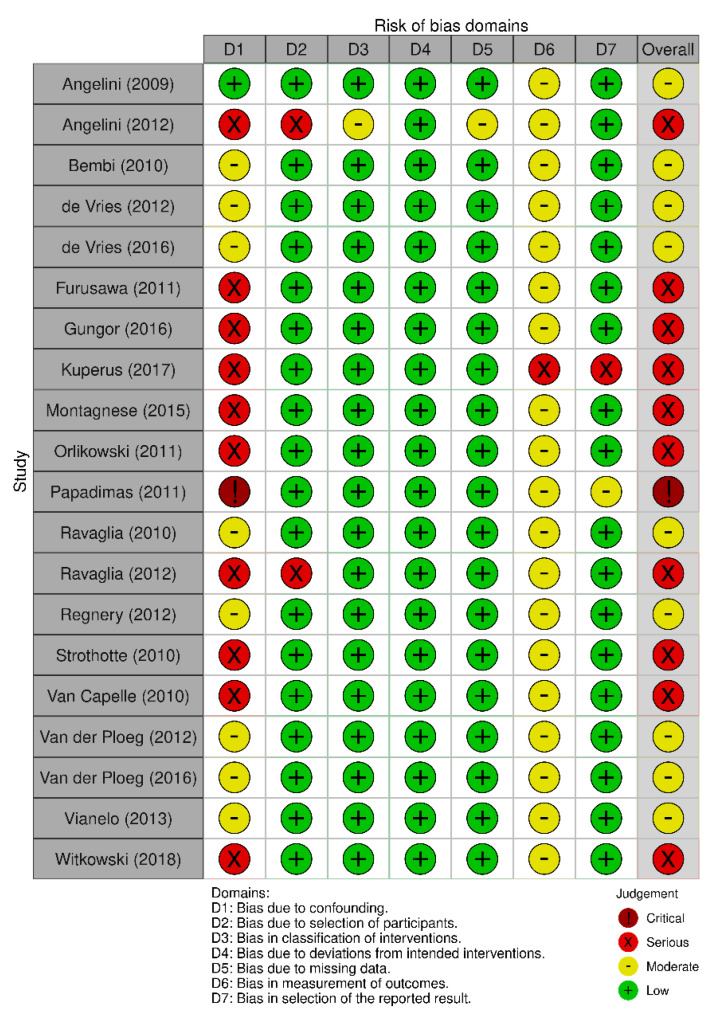
Risk of bias of included studies evaluated through the ROBIN-I tool.

**Table 1 jcm-10-04828-t001:** Database search queries.

Database	Search Query
MEDLINE (via PubMed)	“Glycogen Storage Disease Type II” [Mesh] AND “alpha-Glucosidases” [Mesh] AND “humans” [MeSH]
Embase	“glycogen storage disease type 2”/exp AND “recombinant glucan 1, 4 alpha glucosidase”/exp OR “recombinant glucan 1,4 alpha glucosidase”

**Table 2 jcm-10-04828-t002:** Outcomes of interest defined a priori and studies that met the inclusion criteria.

Outcome	Number of Articles	References
Assessment of functional capacity:		
-FVC	15	[13,27,28,29,30,31,32,33,34,35,36,37,38,39,40]
-6MWT	14	[13,27,28,29,30,31,34,35,36,37,38,39,41,42]
-WGMS	6	[13,27,30,31,40,43]
Safety	14	[13,28,29,30,31,33,34,36,39,40,42,43,44,45]
Upper-limb strength	9	[13,28,29,30,33,34,36,39,46]
Quality of life	6	[13,28,30,36,43,47]
Time on ventilation	6	[30,31,33,40,42,43]
Survival	0	-
Sleep quality	0	-
Swallowing disorder	0	-

FVC = forced vital capacity. 6MWT = 6-min walking test. WGMS = Walton and Gardner-Medwin Scale.

**Table 3 jcm-10-04828-t003:** Included studies and their characteristics.

Author	Patients (*n*/Male)	Design	Age at Onset of ERT—yo–μ (sd) (Range)	Follow-Up Duration	Control	Patients on Ventilation (*n*)
Angelini et al. (2009) [37]	11/3	Cohort	31.1 (8)	* N/A	-	1/11
Angelini et al. (2012) [31]	68/33	Cohort	43 (15.4) (7 to 72)	36 months	-	27/68
Bembi et al. (2010) [42]	24/14	NRSI	Young: 12 (3.3) Adults: 47.6 (10.7)	36 months	-	9/24
de Vries et al. (2012) [33]	49/21	Cohort	52.1 (median) (26.2 to 76.3)	23 months	-	13/49
de Vries et al. (2017) [45]	73/37	NRSI	52 (26 to 74)	36 months	-	22/73
Forsha et al. (2011) [44]	87/44	Post-hoc analysis of RCT	44 (39 to 52)	19.5 months	Placebo	N/A
Furusawa et al. (2011) [32]	5/2	Case series	47 (13.6) (32 to 66)	24 months	-	5/5
Gungor et al. (2016) [47]	174/81	Cohort	50 (median) (24 to 76)	*120 months	-	84/174
Kuperus et al. (2017) [39]	88/45	Cohort	52 (median) (24 to 76)	73.2 months (median)	-	21/88
Montagnese et al. (2015) [27]	14/N/A	Cohort	53.2 (11.1) (36 to 72)	31 months (mean)	-	N/A
Orlikowski et al. (2011) [43]	5/2	NRSI	47.8 (14.4) (28 to 62)	12 months	-	5/5
Papadimas et al. (2011) [46]	5/1	Cohort	46.8 (14.4) (40 to 73)	12 months	-	N/A
Ravaglia et al. (2010) [35]	11/6	NRSI	54.2 (11.2)	at least 24 months	-	N/A
Ravaglia et al. (2012) [41]	16/7	NRSI	54.5 (15.1)	at least 24 months	-	N/A
Regnery et al. (2012) [30]	38/18	NRSI	53.1 (27 to 73)	36 months	-	13/38
Strothotte et al. (2010) [13]	44/24	NRSI	48.9 (12.9) (21 to 69)	12 months	-	16/44
van Capelle et al. (2010) [34]	5/3	Phase II open study, followed by an extension period	11.1 (3.7) (5.9 to 15.2)	36 months	-	1/5
van der Ploeg et al. (2010) [36]	90/45	RCT (LOTS)	45.3 (12.4) (15.9 to 70)	19.5 months	Placebo	ERT = 20/60 Placebo = 11/30
van der Ploeg et al. (2012) [29]	60/34	Open study (LOTS extension)	45.3 (12.4) (15.9 to 79)	26 months	-	20/60
van der Ploeg et al. (2016) [28]	16/7	NRSI	51.6 (13.7) (24.5 to 70.7)	6 months	-	0/16
Vianello et al. (2013) [40]	Group A: 8/5 Group B: 6/1	Cohort with historical control	Group A: 51.5 (12.2) (29 to 65) Group B: 43.8 (15.8) (18 to 59)	Group A = 35.8 months (mean)	Group B (Historical control without ERT) = 52.6 months (mean)	Group A=8/8 Group B=6/6
Witkowski et al. (2018) [38]	5/2	Case series	35.8 (26 to 41)	72 months	-	N/A
TOTAL	896/388	-	42.8 (7 to 72.3)	32.5 months	-	265

NRSI = non-randomized study of interventions. N/A = not available. LOTS = late-onset treatment study. All studies used alglucosidase alfa IV 20 mg/kg/biweekly, except those marked with *, in which dosage was not specified.

**Table 4 jcm-10-04828-t004:** Summary estimates for incidence of safety outcomes of enzyme replacement therapy for patients with late-onset Pompe disease.

					Summary IR (95% CI)	
Outcome	Studies	Participants	P_Q_	I^2^	Random-Effects Model	Fixed-Effects Model
Mortality	9	675	0.66	38.9	0.44 (0.15 to 1.28)	0.56 (0.31 to 1.01)
AB+	7	323	<0.001	94.7	42.63 (24.07 to 75.49)	35.28 (31.41 to 39.62)
AE	3	139	<0.001	97.4	30.93 (2.96 to 323.51)	26.59 (21.0 to 33.67)
SAE	5	367	<0.001	89.7	4.19 (0.63 to 27.69)	2.32 (1.52 to 3.57)
IAR	4	43	<0.001	97.2	3.03 (0.03 to 305.58)	23.71 (16.26 to 34.58)
Patients with IAR	7	274	<0.001	94.1	6.58 (1.67 to 25.93)	6.69 (5.20 to 8.60)

AE = adverse event. SAE = serious adverse event. IAR = infusion-associated reaction. AB+ = presence of anti-alglucosidase alfa antibodies. P_Q_ denotes the *p*-value for Cochran’s Q test. I^2^ is the fraction of variance that is due to statistical heterogeneity and not chance. 95% CI = 95% confidence interval. IR = incidence rate (expressed in number of events per 100 person-years).

**Table 5 jcm-10-04828-t005:** Antibody titers for patients with late-onset Pompe disease receiving enzyme replacement therapy, reduction in response to treatment, and incidence of AEs.

Study	N of Patients	N Ab Titer ≥ 1:250	Method of Measuring Ab	Reduced Response to Treatment	AEs Attributable to Ab Presence
Angelini et al. (2012) [31]	15	11	N/A	N/E	Yes (*n* = 1)
de Vries et al. (2017) [45]	73	46	Van Gelder et al. (2014) [48]	Yes (*n* = 1)	Yes
Kuperus et al. (2017) [39]	73	44	Van Gelder et al. (2014) [48]	Yes (*n* = 1)	N/E
Orlikowski et al. (2011) [43]	5	5	N/A	No	N/E
Regnery et al. (2012) [30]	38	38	N/A	Yes (*n* = 1)	N/E
van Capelle et al. (2010) [34]	5	5	N/A	No	N/E
Van der Ploeg et al. (2010) [36]	59	59	Kishnani et al. (2006) [49]	N/E	No
Van der Ploeg et al. (2012) [29]	59	59	Kishnani et al. (2006) [49]	Yes (*n* = 2)	No

AE = adverse event. Ab = anti-alglucosidase alfa antibodies (measured by different methods). N/A = not available. N/E = not evaluated.

## Data Availability

The datasets analyzed during the current study are available from the corresponding author on reasonable request.

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
