# Peer review of "A Systematic Review and Meta-Analysis of Enzyme Replacement Therapy in Late-Onset Pompe Disease"

_jcm, 2021, doi:10.3390/jcm10214828_

Round 1

Reviewer 1 Report

 Please next time add a clean copy of the manuscript at the end of the changes in the manuscript, it would make it easier to read and follow your corrections.

Author Response

Answers to the Reviewer #1

1) Please next time add a clean copy of the manuscript at the end of the changes in the manuscript, it would make it easier to read and follow your corrections.

R: Thank you. We appologize for the inconvenience. We submitted a .pdf version clean of changes. 

Reviewer 2 Report

Name of the Journal: Journal of Clinical Medicine

Manuscript Draft

Manuscript Number: jcm-1384124

Title: systematic review and meta-analysis of enzyme replacement therapy in late-onset Pompe disease.

Article Type: systematic review and meta-analysis.

Authors: Alícia Dorneles Dornelles ; Ana Paula Pedroso Junges ; Tiago Veiga Pereira ; Bárbara Corrêa Krug ; Candice Beatriz Treter Gonçalves ; Juan Llerena Jr ; Priya Sunil Kishnani ; Haliton Alves de Oliveira Junior ; Ida Vanessa Doederlein Schwartz.

Synthesis

Dorneles Dornelles et al. performed a well-designed review and meta-analysis addressing a very interesting question. In that scope the aim is well and clearly stated. The method is rigorous and appropriate. The results are clearly presented and straightforward. The conclusion a well presented. The authors conclude that ERT in LOPD lead to clinical relevant and statistically significant benefit for three items being 6MWT, TOV and QoL measured by the SF-36 Physical component. Interestingly, the cannot confirm a relevant benefit concerning potential stabilization or improvement about the FVC representing the respiratory function of these patient that is probably the most important parameter linked to the mortality in this disease and also related with the disability of these patients.   

Global assessment

Overall this is a very good paper that merits a publication with a high rating in a journal such as Journal of Clinical Medicine.

Major concerns: none

Minor concerns: see below.

Title: no comments.

Highlights: no comments.

Abstract: no comments.

Abbreviations: please provide systematically in the text for all abbreviation, when used for the first time, the in extensor version followed by the abbreviation between parentheses.

Introduction: No comments

Methods: No comments

Results:

  • In Figures 2 to 7: please add the “n” for each studies as well as the reference numbering (in order to avoid any confusion).
  • In Figure 2 and 3 why the LOTS study (Van de Ploeg 2010) is not included since this is RCT with primary outcomes being FVC and 6MWT ?
  • In table 4, please detail in the legend what is I². Moreover the legend mention IR that is not clearly present in the table, please correct it.
  • Page 8/30 line 380: “correspoto” ??? please correct it.

Discussion: No comments

Conclusions: No comments

The numbering of the pages in fantasist and need be corrected.

References: no comments.  

Photography: not applicable.

Legend: no comments.  

Language: The paper is well written, no particular comment.

The references are complete and accurate.       

Ethical issues: not applicable.

Author Response

Dorneles Dornelles et al. performed a well-designed review and meta-analysis addressing a very interesting question. In that scope the aim is well and clearly stated. The method is rigorous and appropriate. The results are clearly presented and straightforward. The conclusion a well presented. The authors conclude that ERT in LOPD lead to clinical relevant and statistically significant benefit for three items being 6MWT, TOV and QoL measured by the SF-36 Physical component. Interestingly, the cannot confirm a relevant benefit concerning potential stabilization or improvement about the FVC representing the respiratory function of these patient that is probably the most important parameter linked to the mortality in this disease and also related with the disability of these patients.

Overall this is a very good paper that merits a publication with a high rating in a journal such as Journal of Clinical Medicine.

1) Abbreviations: please provide systematically in the text for all abbreviation, when used for the first time, the in extensor version followed by the abbreviation between parentheses.

R: Thank you. We reviewed all abbreviations and corrected the necessary ones.

2) In Figures 2 to 7: please add the “n” for each studies as well as the reference numbering (in order to avoid any confusion).

R: Thank you. We added the requested information.

3) In Figure 2 and 3 why the LOTS study (Van de Ploeg 2010) is not included since this is RCT with primary outcomes being FVC and 6MWT ?

R: Thank you for your question. As stated in methods and defined a priori, line 173-175, “Studies with overlapping data were excluded from meta-analyses. In these cases, the study with the largest sample (or, if both studies had the same sample size, that with the longest follow-up) was retained for analysis.”. Therefore, we included LOTS Extension trial in the meta-analysis (Van der Ploeg et al., 2012).

4) In table 4, please detail in the legend what is I². Moreover the legend mention IR that is not clearly present in the table, please correct it.

R: Thank you. We added an explanation for I². For the IR issue, we adjusted the table for fitting all in the same page, in order to avoid misunderstandings.

5) Page 8/30 line 380: “correspoto” ??? please correct it.

R: Thank you. The sentence was corrected to “the first, of 16 patients, corresponded to high titers (>1: 31,250), the second, of 29 patients, corresponded to moderate titers (1: 1250 to <1: 31,250); the third, of 28 patients, to low titers (0 to <1: 1250).”.

6) The numbering of the pages in fantasist and need be corrected.

R: Thank you. We corrected it.

This manuscript is a resubmission of an earlier submission. The following is a list of the peer review reports and author responses from that submission.

Round 1

Reviewer 1 Report

It would be informative if the authors include calculated p values for the characters that changed significantly by ERT in the abstract and in the discussion.

Please include an abbreviation list of the terminologies and the statistical methods that you used in this study, probably at the beginning of the manuscript. There are several abbreviations that have not be defined.

Please define “t-values” on page 5, second paragraph.

In the 22 studies, if there is any evaluation and data about life expectancy changes in patients under ERT treatment please include the data.

Regarding heterogeneity evaluation, on the method section, the Cochran’s Q was mentioned as the method for statistical heterogeneity evaluation.  Please describe on Figure 2-7 the T2 and I2 calculations.

Please define SF-36 questionnaire.

Perhaps the columns “Weight” on figures 2-7 are related to Cochran’s Q test, please define it in the methods or their figure legends.

Please try to summarize the data related to “Anti-alpha-algucosidase antibodies” into a table.

Are there any information about the genetic background, population, and ethnicity of each cohorts in this study?  It would be informative to evaluate the effect of each population in this study, if it is possible.

Reviewer 2 Report

The efficacy or ERT in LOPD it is addressed by several papers over the last years and some systematic reviews have be published as mentioned also by the authors. Data on long term effects of ERT are poor. In the present paper the authors revised all the prospective studies on ERT efficacy and claim to concentrate their analysis on two aspects that can be of interest : variation of hours of ventilation (TOV) and quality of life ( QoL). 

The studies selected according to their inclusion criteria were not too much and moreover the studies where the above aspects are taken into  consideration are few. For QoL just 3 studies were considered whereas for TOV  4 studies. A critical point is also related to the time of observation in these studies that is quite heterogeneous and not comparable . For instance Gungor et al data are over 10 years whereas Orkikonskwi was only 12 months. 

It is known that ERT efficacy decline after 2 to 3 years from ERT start  ( Haarlar et al. ) and this should be considered.

I suggest to raise these points in the discussion and comment on these.